# A Comparative Characterization and Expression Profiling Analysis of *Fructokinase* and *Fructokinase-like* Genes: Exploring Their Roles in Cucumber Development and Chlorophyll Biosynthesis

**DOI:** 10.3390/ijms232214260

**Published:** 2022-11-17

**Authors:** Lianxue Fan, Wenshuo Zhang, Zhuo Xu, Shengnan Li, Dong Liu, Lili Wang, Xiuyan Zhou

**Affiliations:** 1Key Laboratory of Biology and Genetic Improvement of Horticultural Crops (Northeast Region), Ministry of Agriculture, College of Horticulture and Landscape Architecture, Northeast Agricultural University, Harbin 150030, China; 2College of Advanced Agriculture and Ecological Environment, Heilongjiang University, Harbin 150080, China; 3Division of Plant Protection, College of Agriculture, Northeast Agricultural University, Harbin 150030, China

**Keywords:** cucumber, *FRK*, *FLN*, gene expression, sink tissues, chlorophyll biosynthesis

## Abstract

Fructokinase (FRK) and fructokinase-like (FLN), belonging to the phosphofructokinase B type subfamily, share substantial sequence similarity, and are crucial in various plant physiological processes. However, there is limited information regarding what functionally differentiates plant FRKs from FLNs. Here, a total of three *CsFRK*s and two *CsFLN*s were identified from the cucumber genome. Their significant difference lay in the structure of their G/AXGD motif, which existed as GAGD in *CsFRK*s, but as G/ASGD in *CsFLN*s. Comparative phylogenetic analysis classified *CsFRK*s and *CsFLN*s into five sub-branches consistent with their quite different exon/intron organizations. Both transcriptome data and RT-qPCR analyses revealed that *CsFRK3* was the most active gene, with the highest expression in the majority of tissues tested. Moreover, the expression levels of two putative plastidic genes, *CsFRK1* and *CsFLN2*, were significantly positively associated with chlorophyll accumulation in the chlorophyll-reduced cucumber mutant. Briefly, both *CsFRK* and *CsFLN* genes were involved in the development of sink tissues, especially *CsFRK3*. *CsFRK1* and *CsFLN2* were recognized as candidates in the chlorophyll biosynthesis pathway of cucumber. These results would greatly assist in further investigation on functional characterization of *FRK*s and *FLN*s, especially in the development and chlorophyll biosynthesis of cucumber.

## 1. Introduction

Fructose is one of the dominant sugars in cucumber (*Cucumis sativus* L.) fruit [1] and is derived from sucrose. Several sugar transporters and other enzymes related to sugar allocation and partitioning have been functionally characterized, such as *CsSWEET7a*, *CsSUS3*, and *CsSUS4* [2,3,4]. However, for fructose metabolism only, the systematic and functional identification of the key enzyme genes is still lacking in cucumber, including fructokinase (FRK) and fructokinase-like (FLN) family genes.

FRKs belong to the phosphofructokinase B type (pfkB) family and are crucial catalytic enzymes involved in the phosphorylation of fructose [5,6]. Except for two signature motifs (a di-GLY (GG) motif in the N-terminal region and a G/AXGD motif in the C-terminal region) from pfkB family, FRK proteins also possess several unique and conserved features, including an ATP-binding and a substrate-binding motif [7,8]. Another pfkB family of proteins, FLNs are special, sharing substantial sequence similarity with known FRKs, yet they do not possess any FRK activity. The significant difference between FRKs and FLNs is in the structure of their G/AXGD motif, which exists as GAGD in FRK while as GSGD or A/QSGD in FLN [9,10,11]. However, the functional differences between plant FRKs from FLNs have not been extensively studied.

The FRK family has been best characterized functionally and biochemically in a few plant species, especially *Arabidopsis* and tomato [5,12]. *Arabidopsis frk6* mutants show slightly delayed flowering under short-day conditions, which is similar to the function of *SlFRK1* [13,14]. *Arabidopsis frk6 frk7* double-mutant exhibited normal growth in soil but yielded dark, distorted seeds, and the seed distortion can be complemented by over-expression of *SlFRK1* [15]. Tomato *SlFRK2* and *SlFRK3* together are required for xylem fiber development, and the plastidic *SlFRK3* gene is essential for proper xylem development and hydraulic conductance [16]. *SlFRK4* is activated throughout the later stages of anther development and is co-expressed in mature, germinated pollen together with the invertase LIN7 [17]. Additionally, several *FLN* genes are considered regulators related to the process of plastid transcription. For instance, a rice WLP2 belonging to pfkB family and its paralog OsFLN2 can physically interact with thioredoxin OsTRXz to form a TRX-FLN regulatory module that not only regulates the transcription of the genes encoded by plastid-encoded RNA polymerase but also maintains the redox balance in chloroplasts under heat stress [18]. In a rice mutant *rey(k2)*, the disruption of PFKB1 protein led to leaf yellowing, and PFKB1 functioned in early chloroplast development, partly by regulating chloroplast-associated genes [19]. Similarly, AtFLN1, AtFLN2 and another a pfkB protein NARA5 are indispensable for the hyperexpression of photosynthetic genes encoded in the plastid genome [20,21]. Both *FRK*s and *FLN*s are not only involved in sink-source interaction but also in the regulation of plant growth and chloroplast development.

In this work, at least three cucumber *FRK* and two *FLN* genes were identified at the whole genome level, and their differences were further analyzed in several aspects, including phylogenetic classification, protein structure, exon/intron organization and promoter *cis*-element. Additionally, their expression patterns of *CsFRK*s and *CsFLN*s were compared in different tissues and chlorophyll-reduced mutant to explore their possible functional roles. These findings will provide a better understanding of the basic characteristics and the further functional characterization of FRK and FLN family members in cucumber.

## 2. Results

### 2.1. Identification of CsFRKs and CsFLNs at the Whole-Genome Level

Using the HAMMER-based method integrating with SMART and InterPro analyses, a total of 15 cucumber PfkB family members were identified from the cucumber genome database. To further determine which cucumber PfkB enzymes belong to FRKs and FLNs, a comparative phylogenetic tree of the 15 putative cucumber and 22 *Arabidopsis* pfkB proteins was constructed. As shown in Figure 1, CsaV3_3G021590.1, CsaV3_5G005580.1 and CsaV3_6G006740.1 were clustered into the clade containing seven active *Arabidopsis* FRKs, named *CsFRK*1–3. CsaV3_3G034260.1 and CsaV3_4G031430.1 belonged to the FLN clade, called *CsFLN*1–2. The rest of CsPfkB members were excluded because they were assigned to other branches (Figure 1a, Appendix A).

Multiple sequence alignment showed that two signature motifs of PfkB domain, a GG and a G/AXGD motif, were found in the amino acid sequences of *CsFRK*s and *CsFLN*s. Notably, the G/AXGD motif was in the form GAGD in *CsFRK*s, while G/ASGD in *CsFLN*s, respectively (Figure 1b). Additionally, the *CsFRK* and *CsFLN* proteins had two conserved domains including the substrate binding and ATP-binding domains (Figure 1b). These results also confirmed the identities of *CsFRK*s and *CsFLN*s. Besides, *CsFRK*s ranged from 331 to 384 amino acids in length, while *CsFLN*1 and *CsFLN*2 were 511 and 588 amino acids long, respectively. The molecular weights of *CsFLN*s (58.11 kDa and 65.90 kDa) were higher than those of *CsFRK*s (35.64 kDa–41.18 kDa), and the same was true of the theoretical isoelectric points (pIs) (Appendix A).

### 2.2. Gene Distribution and Duplication Analyses of CsFRKs and CsFLNs

Based on chromosome (Chr) location information, *CsFRK1*, *CsFRK2* and *CsFRK3* were distributed in Chr3, Chr5 and Chr6, while *CsFLN1* and *CsFLN2* were on Chr3 and Chr4, respectively (Appendix A). In gene duplication analysis, four pairs of dispersed duplicates were determined: *CsFRK1*–*CsFRK3*, *CsFRK2*–*CsFLN1*, *CsFLN1*–*CsFRK3* and *CsFLN2*–*CsFRK3*, and three DNA-transposed duplications were identified, *CsFRK1*–*CsFRK2*, *CsFRK2*–*CsFRK3* and *CsFLN1*–*CsFLN2* (Appendix A). Hence, dispersed and DNA-transposed duplications might contribute to cucumber *FRK* and *FLN* gene family expansion and evolution.

### 2.3. Phylogenetic Relationships of the FRKs and FLNs from Cucumber and Other Plants

To elucidate the phylogenetic relationship, the 30 amino acid sequences encoded by the *FRK* and *FLN* genes from 7 plant species including 5 monocotyledons (cucumber, *Arabidopsis*, tomato, sugarbeet, potato) and 2 dicotyledons (rice and maize) were used to construct a comparative neighbor-joining (NJ) tree. According to the NJ tree, each sub-branch contained FRKs or FLN from both monocotyledons and dicotyledons (Figure 2a), suggesting that these classes originated before the monocot-dicot split and shared the same ancestor. FLNs and FRKs were clearly distinguished, and all FRKs and FLNs were further grouped into six sub-branches, referred to as Class I–Class VI. In detail, *CsFRK*1, *CsFRK*2, *CsFRK*3 were divided into Class II, Class I and Class IV, respectively. *CsFLN*1 and *CsFLN*2 were assigned into Class V and Class VI (Figure 2b), which should be because of their G/AXGD motifs as shown in Figure 1b.

In *Arabidopsis*, AtFRK3 is a plastidic FRK, while the other six FRKs (AtFRK1–2, and AtFRK4–7) are observed in the cytosol [12]. Tomato SlFRK3 is located in the chloroplast, and SlFRK1, SlFRK2 and SlFRK4 are cytosolic enzymes [22]. Two *Arabidopsis* AtFLNs were found in the chloroplast stroma [10], and rice OsFLN2 is confirmed to contain chloroplast transitpeptide [23]. Herein, the subcellular localization of the other nine FRKs and seven FLNs in the NJ tree were predicted by WoLF PSORT program. Among the FRK proteins, the 15 members from Class I, III and IV were predicted to be cytosol-localized except *CsFRK*3 and StFRK, whereas the ones from Class II were plastidic proteins including *CsFRK*1, AtFRK3 and SlFRK3 (Figure 2c). Two *CsFLN*s, unlike the other eight FLNs, were uniformly located in the chloroplasts (Figure 2c), indicating that their functions were relatively conservative.

A syntenic analysis of FRKs and FLNs was conducted between cucumber and *Arabidopsis*. The syntenic map showed that only two pairs of syntenic orthologous genes were identified, *CsFRK2*–*AtFRK1* and *CsFLN1*–*AtFLN1*, indicating that they share a common ancestor with their counterparts (Figure 3). This result was consistent with the classification of *CsFRK*s and *CsFLN*s, which might serve as their phylogeny-based functional prediction.

### 2.4. Conserved Motif and Gene Structure Analyses of the FRKs and FLNs in Cucumber and Other Plants

A total of 10 conserved motifs were identified from all the FRK and FLN proteins. Motifs 1, 2, 4, 5, 6 and 7 were shared by all FRKs and FLNs. Among them, Motif 2 contained the GG motif in the N-terminal region, and Motif 7 represented the G/AXGD motif in the C-terminal region, which were the basic regions of pfkB domain. Motif 3 only appeared in all FRKs, and motifs 8 and 9 appeared in all FLNs, but motif 10 only appeared on the FLN1s in Class V and OsFLN2 in Class VI (Figure 4a,b). In any case, the motif compositions of whether plant FRKs or FLNs were evolutionarily conserved.

Exon/intron organization analysis showed that the most *FRK*s and *FLN*s from the same branch possessed the same number of exons, though they were variable in both gene length. For example, the FRKs from Class I and II had seven exons with the largest number, and those of Class IV contained four exons. FLN1s in Class V and FLN2s in Class VI were detected to have two and five exons, respectively (Figure 4a,c). The exon/intron organizations might imply diverse functions of the FRK and FLN family, which seemed to better support the group classification than motif compositions.

### 2.5. Cis-elements in the Promoter Region of CsFRKs and CsFLNs

The promoter sequences (2000 bp) of *CsFRK*s and *CsFLN*s were uploaded to the PlantCare database to search *cis*-acting elements, and four main *cis*-acting elements were collectively identified in the promoter regions of these five genes, involved in light responsive, plant hormone, stress responsive and transcription factor (TF) (Figure 5a). In particular, two meristem-specificity element CAT-boxes were found in the *CsFLN1* promoter, and only one GCN4 motif was identified in the *CsFRK3* promoter, which functions as an essential *cis*-element for endosperm-specific gene expression. Additionally, both MYB- and MYC-responsive elements exited in five *CsFRK* promoters, indicating that MYB and MYC TFs were putative transcriptional regulators of *CsFRK* genes (Figure 5b). These specific *cis*-acting elements might contribute to the diverse functions of the *FRK* and *FLN* gene family in cucumber.

### 2.6. Expression Patterns of CsFRKs and CsFLNs in Different Tissues

The expression of *FRK* genes is detected mainly in plant sink tissues such as roots, stems, flowers, fruits and seeds [11]. Here, the published RNA-seq data of three typical sink organs (pedicle, stalk and fruit) in cucumber were used to investigate the tissue-specific expression levels of *CsFRK*s and *CsFLN*s. As showed in Figure 6a, each gene showed higher transcription levels in the pedicle, stalk and fruit, with transcripts per ten million (TPTM) values varying between 67–1335.5 (Figure 6a), indicating they might play stabilizing roles in the three sink tissues. Among them, *CsFRK3* was more abundantly expressed; for instance, its expression levels in pedicle were 6.20, 6.28, 5.08 and 14.99 times higher than those of *CsFRK1*, *CsFRK2*, *CsFLN1* and *CsFLN2*, respectively. *CsFLN1* exhibited higher expression levels than those of *CsFLN2* in the three sink tissues.

Otherwise, the patterns of *CsFRK* and *CsFLN* genes expression were further examined in five different tissues of cucumber planta by real-time quantitative polymerase chain reaction (RT-qPCR) analysis, including leaf, stem, tendril, female flower and male flower. The results showed that all *CsFRK* and *CsFLN* genes had diverse expression patterns. *CsFRK3* were expressed at higher levels than that of the rest in most tissues. Specifically, *CsFRK3* was in female flower with high expression (Figure 6b), suggesting that it might be involved in female cucumber flower development.

### 2.7. Potential Roles of CsFRKs and CsFLNs in Chlorophyll Synthesis Pathway

Currently, several pfkB family members have been reported to be involved in the process of chloroplast development in leaf color mutants, especially the plastid-localized proteins [18,19]. To verify whether the cucumber FRK or FLN family members have similar functions, a cucumber yellow leaf mutant (Y) and its wild type (G) were used in this study. The Y line displayed a yellow color throughout its whole life, along with less total chlorophyll content. According to the results of RT-qPCR analysis, the mRNA expression of all *CsFRK*s and *CsFLN*s exhibited significant differences between the function leaves of the G and Y. Among them, *CsFRK1* and *CsFLN1* were upregulated, while *CsFRK2*, *CsFRK3* and *CsFLN2* were downregulated in the yellow function leaves compared with that in green ones (Figure 7a).

Moreover, the positive correlations were found between the expression levels of *CsFRK1* and *CsFLN2* and the total chlorophyll contents of various tissues, such as young leaf (Yl), stem (St), tendril (Te), peel (Pl) and flesh (Fe) (*CsFRK1*, r = 0.9456, *p* < 0.0001; *CsFLN2*, r = 0.8515, *p* = 0.0018). Linear regression analysis showed that their intimate relationships could explain 89.42% and 72.50% of the variation of chlorophyll content in cucumber, respectively (Figure 7b–d). Therefore, both *CsFRK1* and *CsFLN2* were recognized as the potential participants in the pathway of chlorophyll synthesis in cucumber.

## 3. Discussion

Plant FRKs are enzymes with a high, specific affinity for fructose and also the phosphorylating enzymes necessary for the conversion of fructose to fructose-6-phosphate [11]. To date, plant FRKs have been assigned to the pfkB subfamily based on their sequence similarity. Several of the pfkB proteins that share substantial sequence similarity with known FRKs are found in plants but do not possess any FRK activity, rather playing an indispensable role in plant growth and chloroplast development, such as *Arabidopsis AtFLN1* and *AtFLN2* [9,24]. This present study provided some available information about the similarities and differences of cucumber *FRK* and *FLN* genes in their structure and functions.

### 3.1. Basic Characteristics of Cucumber FRKs and FLNs

In the present work, a total of three *CsFRK* and two *CsFLN* genes were identified at the whole genome level of cucumber. Consist with other plant FRKs and FLNs, the significant difference between *CsFRK* and *CsFLN* lay in the structure of their G/AXGD motif. In *CsFRK*s, the G/AXGD motif was GAGD, while in *CsFLN*s it was G/ASGD [9,10,11]. Another significant difference was their amino acids length, which due to FLNs longer N-terminal sequence and at least three additional insertions [11].

Evolutionary analysis indicated that the FRKs and FLNs in cucumber and six other plant species could be divided into six sub-branches that were also supported by their gene structure and conserved motifs, and their gene structures better reflected their variations. Structural differences of *CsFRK*s and *CsFLN*s might contribute to their functional divergence, which embody their various expression patterns as described by this paper. This view was consistent with the results observed in other plants, such as cassava and *Populus* [6,25].

Protein subcellular localization is key characteristic of protein functional research [26]. Interestingly, the genes that clustered together generally possessed the same intracellular locations, illustrating that they not only had similarity in genetic evolution, but also might play analogous roles in cells. The intracellular localization of *Arabidopsis* AtFRK3 and tomato SlFRK3 expressed are visualized in chloroplasts, while their other FRK members are cytosolic enzymes [12,22]. For three cucumber FRKs, only *CsFRK*1 in Class II that contained AtFRK3 and SlFRK3 was predicted to be the plastid-localized proteins. Therefore, it was reasonable that majority of plant species may also have a single plastidic FRK and several cytosolic FRKs, though the accurate subcellular localizations of *CsFRK*s require further experimental confirmation.

In the synteny analysis integrating with evolutionary classification, only *CsFRK2* and *CsFLN1* had good collinearity with *Arabidopsis AtFRK1* and *AtFLN1*. *AtFRK1* has been demonstrated to be important for seed oil accumulation and vascular development, and *AtFLN1* is an essential role for plant growth and chloroplast development in *Arabidopsis* [9,10,15]. Consequently, as their orthologues, *CsFRK2* and *CsFLN1* were presumed to play similar biological function in cucumber.

### 3.2. Functional Roles of CsFRKs and CsFLNs in Different Tissues

Increasing evidence suggests that *FRK*s and *FLN*s are important players in plant growth and development, especially the roles of FRKs in vascular tissues [6,15,16,27]. Based on RNA-seq and RT-qPCR analyses, all *CsFRK*s and *CsFLN*s exhibited constitutive and diverse expression patterns in various tissues of cucumber planta, especially in the sink tissues including pedicle, stalk and fruit. Similarly, several *FRK* genes are detected in most organs of a few plant species, such as cassava [25], loquat [28] and tomato [29]. These results further suggested a broad role for *CsFRK*s and *CsFLN*s in the growth and development of cucumber. Among *CsFRK*s and *CsFLN*s, *CsFRK3* was the most active gene, with the highest expression level in the majority of tissues tested. Its homologous genes *SlFRK2*, *AtFRK2*, *AtFRK6* and *AtFRK7* have been reported to function in vascular, seed and fruit development [15,30,31].

Additionally, *CsFRK3* was specifically and highly expressed in female flower, which might be attributed to the role of the estrogen response element ERE contained in its promoter. This result was correlated with cassava *MeFRK5* and tomato *SlFRK4* genes, and *SlFRK4* has been confirmed to be a pollen-specific expression gene [25,32]. Thus, it could be inferred that *CsFRK3* might be served as a specifical actor in female cucumber flower development.

### 3.3. Two Plastid-Localized CsFRK1 and CsFLN2 Involved in the Biosynthesis of Chlorophyll

Most green plants commonly contain natural pigments such as chlorophyll, carotenoids and lutein, while their leaves show different colors due to the changes in content and proportion of various pigments. Frequently, leaf yellowing is attributed to decreasing chlorophyll content in leaf color mutants [33], in which the roles of PfkB family members seem to be underestimated. In rice *rey* mutant with yellow leaf at early seedling stage, *PfkB1* is recognized as a novel regulator indispensable for early chloroplast development, as well as affect the RNA expression level of chloroplast related genes [19]. Here, RT-qPCR results demonstrated the transcript expression levels of *CsFRK1* and *CsFLN2* were significantly positively associated with chlorophyll accumulation in the five tissues of cucumber yellow leaf mutant. Additionally, eight and three *cis*-elements involved in light response were further found in the promoters of *CsFRK1* and *CsFLN2* genes. Moreover, *Arabidopsis FLN2*, as a homolog of *CsFLN2*, has been reported to act an important function in chloroplast development and metabolic regulation [21]. Therefore, it is reasonable to hypothesize that some plastid-localized *FRK* genes, such as *CsFRK1* and *CsFLN2*, may be potential regulators in the pathway of chlorophyll synthesis or chloroplast development in cucumber.

Additionally, there were multiple MYB and MYC binding sites in the promoters of two latent *CsFRK* genes related to chlorophyll biosynthesis, suggesting that they could be regulated by MYB or MYC TFs. It is necessary to further verify whether *CsFRK1* and *CsFLN2* affect chlorophyll accumulation through the direct regulation of MYB or MYC TFs.

## 4. Materials and Methods

### 4.1. Characterization and Phylogenetic Analysis of CsFRK and CsFLN Family

The protein sequence file of the cucumber 9930 V3 genome was downloaded from Cucurbit Genomics Database (CuGenDB, http://cucurbitgenomics.org/, accessed on 22 June 2022). An HMMER-based search was performed to predict the PfkB proteins using a functional domain (Pfam ID: PF00294). Subsequently, the putative PfkBs were further examined by InterPro (http://www.ebi.ac.uk/interpro/, accessed on 22 June 2022) and SMART (http://smart.embl-heidelberg.de/, accessed on 22 June 2022) programs.

Full-length protein sequences of the 22 *Arabidopsis* PfkB members [12] were obtained from the TAIR database (https://www.arabidopsis.org/, accessed on 22 June 2022). The sequences of the cucumber and *Arabidopsis* pfkB proteins were aligned using MUSCLE. Thereafter, their phylogenetic relationships were evaluated by maximum parsimony (MP) in MEGA X, and by 100 bootstrap replicates for the phylogram [12], to identify the putative cucumber FRK and FLN members. The MP tree was visualized using iTOL v6 (https://itol.embl.de/, accessed on 22 June 2022). Protein length, molecular mass, and theoretical pIs of FRK and FLNs were calculated by ExPASy3 [34].

The amino acid sequences of FRK and FLN family members from two monocotyledons (maize and rice) and five dicotyledons (cucumber, *Arabidopsis*, tomato, cotton and soybean) were obtained from NCBI database (Appendix A). The phylogenetic tree of FRKs from cucumber and other species was also generated by MEGA X using neighbor-joining (NJ) method with 1000 bootstrap replicates [35].

### 4.2. Identification of Protein Subcellular Localization and Conserved Domains

Protein sequence analysis was performed by both Jalview [36] and GeneDoc software [37]. The conserved motifs were identified using the Multiple Em for Motif Elicitation program (http://meme.nbcr.net/meme/cgi-bin/meme.cgi, accessed on 25 June 2022).

The WoLF PSORT server (https://www.genscript.com/wolf-psort.html?src=leftbar, accessed on 25 June 2022) were used for predicting protein subcellular locations.

### 4.3. Analyses of Gene Localization, Structure, Duplication and Synteny Analysis

Gene chromosomal locations and duplication information were retrieved from CuGenDB and Plant Duplicate Gene Database (PlantDGD, http://pdgd.njau.edu.cn:8080/, accessed on 12 July 2022), respectively, and were visualized by TBtools software [38].

Gene structural classification were illustrated through uploading the BED format file of *CsFRK* genes into an online Gene Structure Display Server (http://gsds.gao-lab.org/, accessed on 27 June 2022) tool.

MCScanX software [39] was used for evaluating the collinear blocks of *FRK*s and *FLN*s between in cucumber and *Arabidopsis thaliana*. All these results were visualized in TBtools [38].

### 4.4. Promoter Cis-Element Analysis

The 2000 bp upstream sequences of the start codon of *CsFRK* and *CsFLN* genes were extracted from CuGenDB database, and were submitted to the PlantCARE online database (http://bioinformatics.psb.ugent.be/webtools/plantcare/html/, accessed on 26 June 2022) to identify the putative *cis*-elements. The result was displayed the enrichment of the *cis*-element using a heatmap by TBtools [38].

### 4.5. RNA-seq-Based Expression Profiling

The published RNA-seq data referred to three tissues (Pe, St and Fr) of cucumber plant were acquired from NCBI Gene Expression Omnibus database (GSE62338) [40]. All protein sequences of *CsFRK*s and *CsFLN*s were mapped to the cucumber 9930 V2 genome using the basic local alignment search tool within BioEdit Software. *CsFRK* or *CsFLN* containing 100% identity was considered as the same gene (Appendix A), and their read counts were summarized by the HTSeq-count. The expression level of each gene was computed as the TPTM value. The visualization of gene expression profile was performed by heatmap in TBtools software [38].

### 4.6. Plant Materials and Sampling

The seeds of inbred cucumber line ‘D1008′ were sown in plastic pots containing mixture of isovolumetric soil and substrate, and grown in a growth chamber under 28 °C/18 °C (day/night) and a 16 h day/8 h night cycle. At the two true-leaf stage, cucumber seedlings were transferred to a greenhouse in the Horticulture Experiment Station, Northeast Agricultural University (Harbin, China). Totally, five tissues (leaf, stem, tendril, female flower and male flower) were separately collected at the fruiting stage, and were stored at −80 °C for further use.

To characterize the potential function of *CsFRK*s and *CsFLN*s in chlorophyll biosynthesis, two cucumber inbred lines, ‘Y’ and ‘G’ were used and grown in the greenhouse. Y was a spontaneous yellow leaf mutation found in the G with green leaf color, which had a North China cucumber genetic background. Y was self-pollinated for six generations, which exhibited a stable yellow color of all above-ground organs throughout its life. At the fruiting stage, a total of five different tissues from Y and G lines including function leaf, Yl, St, Te, Pe and Fe were collected separately for total RNA isolation. For each tissue, three independent biological replicates were sampled (each replicate included a pool of five individual plants).

### 4.7. Assay of Photosynthetic Pigment Content

Photosynthetic pigments of five different tissues including Yl, St, Te, Pe and Fe were extracted in a mix solution with 95% ethanol and acetone (v:v = 1:2). The samples were completely immersed in the mixture and protected from light overnight [41]. The extracting solution was measured at absorbances of 645 nm, 663 nm and 470 nm, and the contents of total chlorophyll were calculated by the formula described by Liu et al. [42]. Measurements of all samples were repeated three times.

### 4.8. RNA Isolation and RT-qPCR Analysis

Total RNA from all above-mentioned samples was extracted with RNAprep Pure Plant Kit with DNase I (Tiangen, Beijing, China), and first-strand cDNA was synthesized by HiScript 1st Strand cDNA Synthesis Kit (Vazyme, Nanjing, China) for RT-qPCR analysis. The RT-qPCR of each gene in triplicate was performed by a qTOWER3 System (Analytik Jena, Germany) according to the manufacturer’s instruction. The specific RT-qPCR primers were listed in Appendix A, and *CsEF1α* was used as internal control. Relative transcript level was calculated by the 2^−ΔΔCT^ method.

Pearson correlation analysis between photosynthetic pigment contents and gene expression levels was performed using GraphPad Prism 9 software.

## 5. Conclusions

In summary, at least three *CsFRK* and two *CsFLN* genes from cucumber genome were systematically characterized to explore their structural and functional differences. The structure of G/AXGD motif was different between *CsFRK*s and *CsFLN*s. Both RNA-seq and RT-qPCR analyses revealed cucumber FRK and FLN family members were involved in the development of sink tissues, especially *CsFRK3*. Additionally, *CsFRK3* might have a specific function in the development of female cucumber flower. The evidence provided by correlation analysis showed that *CsFRK1* and *CsFLN2* potentially participated in the chlorophyll biosynthesis pathway of cucumber. The outcomes of this study contribute to understanding the basic characteristics of FRK and FLN family, and explore their specific roles in cucumber development and chlorophyll biosynthesis.

## Figures and Tables

**Figure 1 ijms-23-14260-f001:**
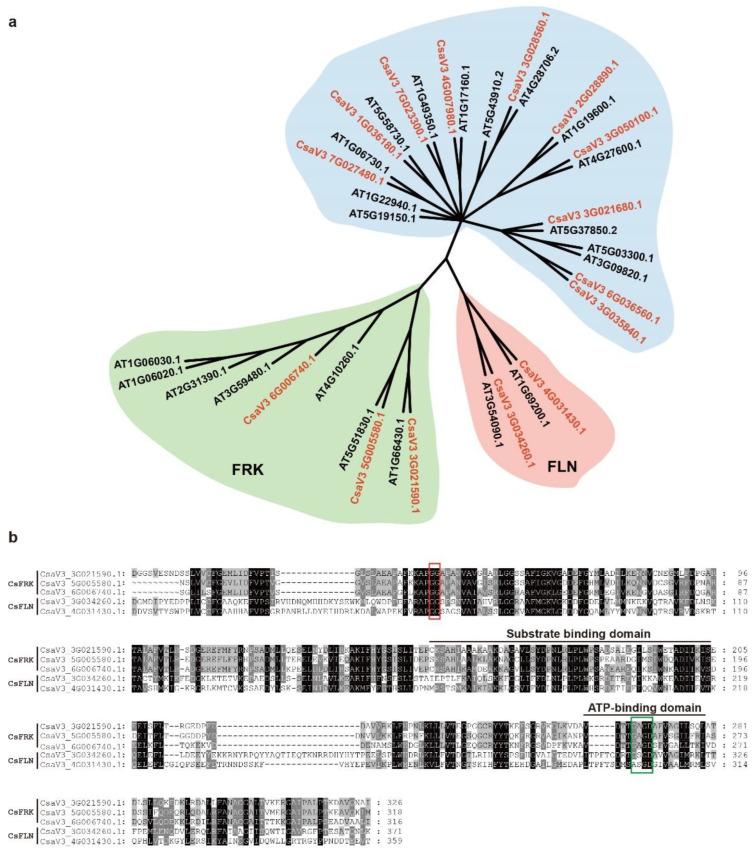
Identification of cucumber FRK and FLN enzymes. (**a**) Consensus bootstrap phylogenetic tree of the pfkB proteins in cucumber (red fonts) and *Arabidopsis* (black fonts). The three cucumber FRKs and two FLNs studied in this manuscript are in green and pink background, respectively. (**b**) Alignment of the putative cucumber FRK and FLN protein sequences. Black and grey boxes present amino acids identical or with conservative substitutions in >50% of the proteins, respectively. The GG and G/AXGD motifs are marked with red and green boxes respectively. The black line is used to indicate the substrate or ATP binding domain.

**Figure 2 ijms-23-14260-f002:**
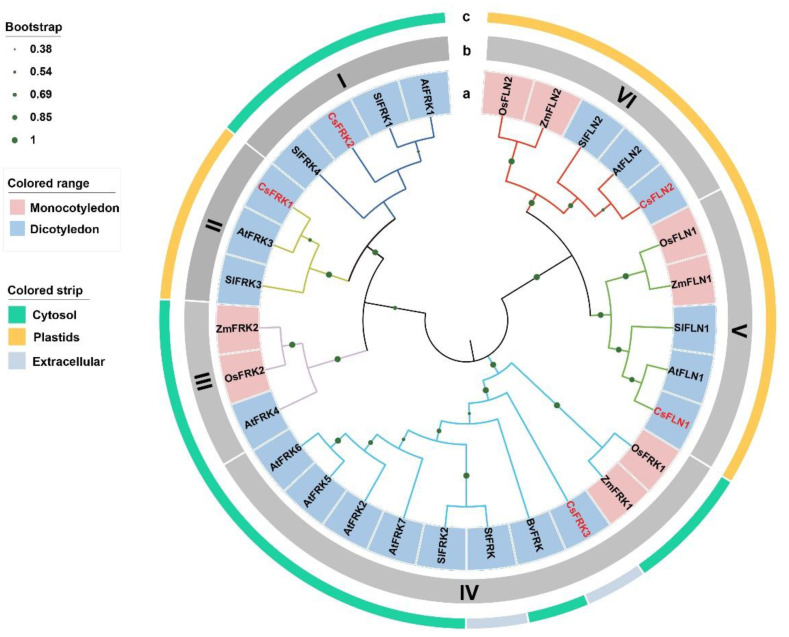
Phylogenetic tree of FRKs and FLNs in cucumber and other plants. (**a**) Evolutionary relationships of FRKs and FLNs in different monocotyledons and dicotyledons. (**b**) All FRKs and FLNs clustered into six sub-branches (Class I–Class VI). (**c**) Subcellular localization of FRKs and FLNs. Five monocotyledons include *Arabidopsis thaliana* (At), *Beta vulgaris* (Bv), *Cucumis sativus* (Cs), *Solanum lycopersicum* (Sl) and *Solanum tuberosum* (St). Two dicotyledons include *Oryza sativa* (Os) and *Zea mays* (Zm).

**Figure 3 ijms-23-14260-f003:**
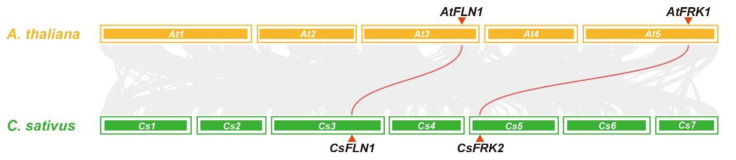
Synteny analysis of *FRK* and *FLN* genes between cucumber and *Arabidopsis*. Colored lines connecting two chromosomal regions indicate syntenic regions between cucumber (*Cs1–7*) and *Arabidopsis* (*At1–5*) chromosomes.

**Figure 4 ijms-23-14260-f004:**
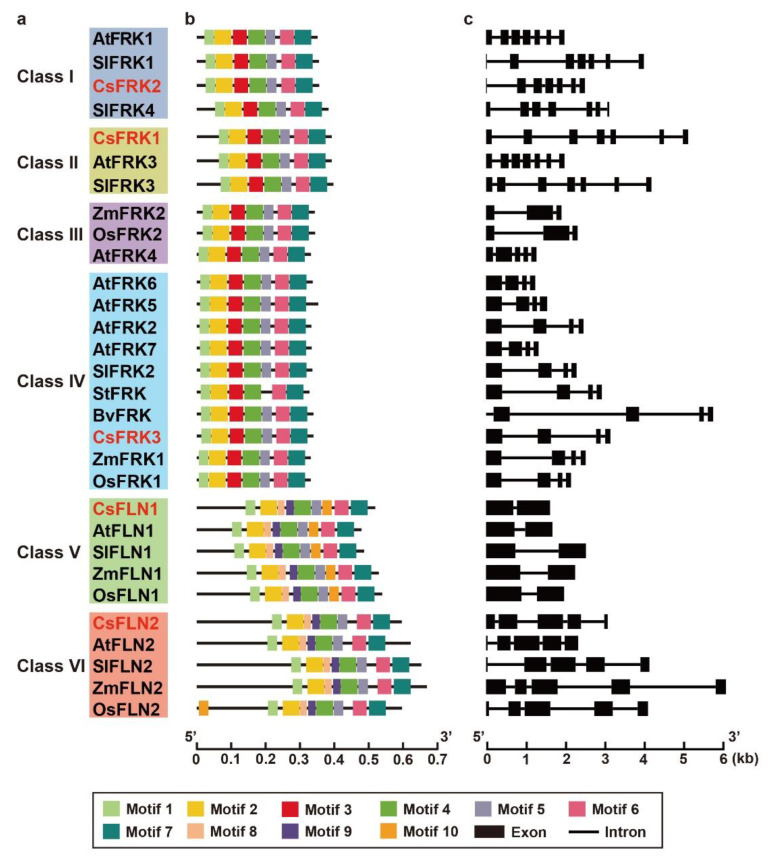
Clustering analysis (**a**), motif architecture (**b**) and exon-intron organization (**c**) of FRKs and FLNs in cucumber and other plants.

**Figure 5 ijms-23-14260-f005:**
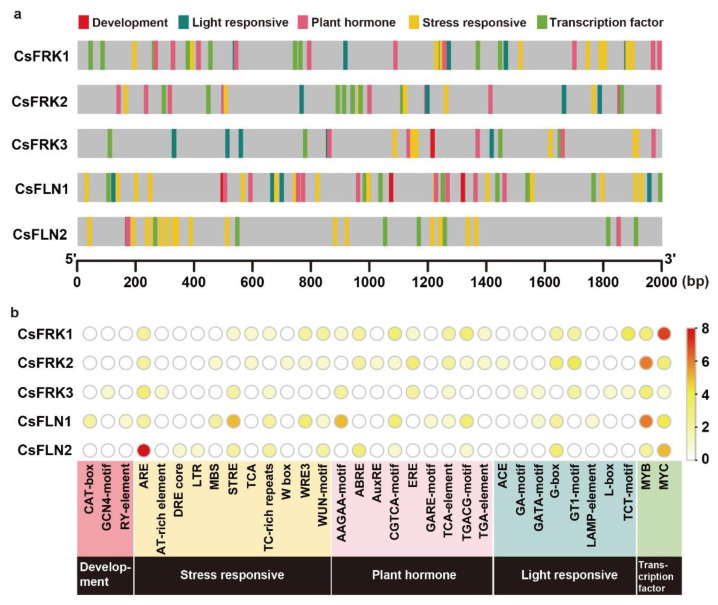
*Cis*-element analysis in promoter of *CsFRK*s and *CsFLN*s. (**a**) Distribution of five main categories of *cis*-element in promoter region. They involved in development, light responsive, plant hormone, stress responsive and transcription factor, respectively. (**b**) Heatmap displaying the number of *cis*-element. ABRE, abscisic acid-responsive element; ARE, antioxidant response element; AuxRE, Auxin responsive element; DRE core, dehydration responsive element core; ERE, estrogen response element; GARE-motif, gibberellin responsive element motif; LTR, low-temperature responsiveness; MBS, MYB binding sites; STRE, stress response element; WRE3, wound response element 3; WUN-motif, wound-motif.

**Figure 6 ijms-23-14260-f006:**
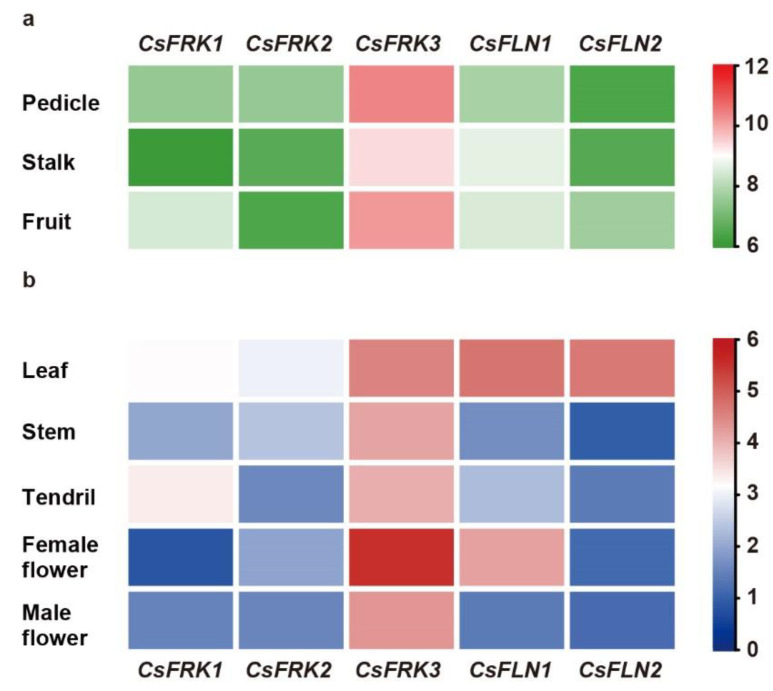
Expression profiles of *CsFRK*s and *CsFLN*s in three sink organs (**a**) and other five tissues (**b**). Heatmap was constructed based on the TPTM and relative expression values, respectively. Colored bar indicates normalized gene expression in log_2_ space.

**Figure 7 ijms-23-14260-f007:**
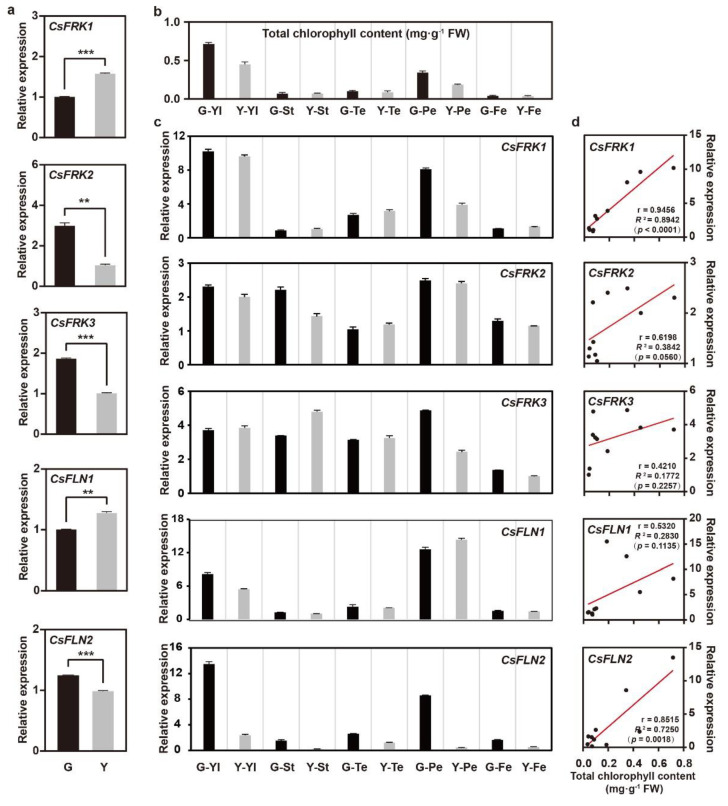
Identification of *CsFRK*s and *CsFLN*s potentially involved in chlorophyll biosynthesis. (**a**) Expression profiles of *CsFRK*s and *CsFLN*s in function leaves of the cucumber yellow leaf mutant (Y) and its wild type (G). Unpaired t-test, *** *p* < 0.0001, ** *p* < 0.01. (**b**) Total chlorophyll contents in five tissues of Y and G plants. Yl, young leaf; St, stem; Te, Tendril; Pe, peel; Fe, flesh. (**c**) Expression profiles of the five genes in different tissues. Error bars indicate standard deviation of three biological replicates. (**d**) Pearson correlation analysis between total chlorophyll content and gene expression level.

## Data Availability

Not applicable.

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
