# Peer review of "A Comparative Characterization and Expression Profiling Analysis of Fructokinase and Fructokinase-like Genes: Exploring Their Roles in Cucumber Development and Chlorophyll Biosynthesis"

_ijms, 2022, doi:10.3390/ijms232214260_

Round 1
Reviewer 1 Report
This study identified three CsFRKs and two CsFLNs from the cucumber genome. Their roles in plant growth, stress tolerance, and chlorophyll biosynthesis in the cucumber were analyzed and determined. First, their structure showed the different G/AXGD motifs, which exist as GAGD in CsFRKs, and G/ASGD in CsFLNs. Comparative phylogenetic analysis classified CsFRKs and CsFLNs into five sub-branches consistent with their different exon/intron organization. Transcriptome data revealed that CsFRK3 exhibited more abundant transcripts in more tissues compared with other CsFRKs or CsFLNs. Under various stresses, most CsFRKs and CsFLNs showed selective differential expression except CsFRK2. Moreover, the expression levels of two putative plastidic genes, CsFRK1 and CsFLN2, were significantly positively associated with chlorophyll accumulation in the different tissues of the cucumber yellow leaf mutant. Briefly, CsFRK and CsFLN family members were involved in developing sink tissues, especially CsFRK3. CsFRK1 and CsFLN2 were recognized as candidates in the chlorophyll biosynthesis pathway of cucumber. These results would greatly assist in further investigating the functional characterization of FRKs and FLNs, especially in plant growth, stress tolerance, and chlorophyll biosynthesis in cucumbers. I suggest this manuscript can be published after minor revisions to its grammar.
1. Line 130-131, "This result indicated that each gene might play its unique action along with the evolution of cucumber.". This conclusion cannot be obtained according to the analysis of the results of the phylogenetic tree, so it is not scientifically rigorous to write this conclusion.
2. Line 157-158, "…, while Motifs 8 and 10 appeared in the FLNs (Figure 3a, b)". From Figure 3b, all FLNs contained motifs 8 and 9, but motif 10 only appeared on the FLNs in Class V and OsFLN2. Please check it.
3. Line 174, "…, four main types of cis-acting elements were collectively identified in the five genes, …". In this sentence, the word "five genes" is not accurate. Please rephrase it accurately.
4. In Figure 4b, all abbreviated words need to be annotated with their full names.
5. Line 204, the full stop after the word "tissues" should be removed.
6. In Figure 6a, the " Y " bar graph in CsFRK5 is missing the error bar.
7. Line 239-240, “(CsFRK1, r = 0.9456, P < 0.0001; CsFLN2, r = 0.8515, P = 0.0018)”, “P” should be lowercase and italicized.
8. For the protein subcellular localization, it is not sure that the results predicted by the software are correct, so it is suggested to analyze additional verification by experiments.
Author Response
Please see the attachment, thank you!

Reviewer 2 Report
The current manuscript investigated functional genomics of fructokinase (FRK) and fructokinase-like (FLN) in cucumber using bioinformatic analyses from data retrieved from public database. The authors also included qPCR and pigmentation content assay to corroborate their results.
I like the way the data have been presented in several figures, which makes it easier for the readers to grasp the content at first look. Good job! Nevertheless, I have major concern and some minor comments about the experimental design and respective analysis and results, which should be addressed before further evaluations.
1) Regarding the RNA-Seq expression analysis topic (Lines 188, Line 373), since the data are from several different studies with various growth conditions and treatments, it is inconclusive to draw a conclusion about sets of genes that can vary under any environment. These are not housekeeping genes that stay unaltered in different treatments. Although finding metadata on plant growth conditions in other studies looks ok, the differences in the experimental time plus the cultivar used is another concern that invalidates the RNASeq analyses. Therefore, due to this major comment, the entire RNASeq content in the paper is not scientifically sound and should be removed.
2) Line 68-73, Should be at the beginning of the introduction section.
3) Line 69, Remove "the".
4) Line 90, replace "was" to "were".
5) Line 101, spell out the abbreviation in the first use; pI.
6) Line 124, spell out ML.
7) Line 134, different colors in figure 2 for the letter "b" has no meaning and are confusing. Please make it unicolor and just keep the letters I-IV.
8) Line 192, spell out Ez.
9) Line 204, replace "." with ",".
10) Line 218, replace "involved" with "involve".
11) Line 221, regarding cucumber yellow leaf mutants, please add a reference or describe in the methods how they have been constructed/obtained.
12) Line 229, figure six has several title errors. In "a", there is no FLN gene. In "b", there is no significant meaning in using a connected scatterplot, instead, use a side-by-side tissue-specific bar chart. Also, there is no "FLN" gene. In "c", what are "NI" and "Fr" in the X axis; the legend description doesn't match. For "d", there is no label in the figure. Also, add FRK and FLN labels as well.
13) Line 239, replace "steam" with "stem".
14) Lines 257-259, please rephrase this sentence to clarify the meaning.
15) Line 276, add the synteny analysis to the result section.
16) Line 318-321, "Therefore, it is reasonable to hypothesize that some plastid-localized FRK genes, such as CsFRK1 and CsFLN2, may be potential regulators in the pathway of chlorophyll synthesis or chloroplast development in cucumber." Although the hypothesis of such statement due to comparative analysis is feasible I am not sure about its reasonability. To be reasonable to mention this statement, a wet-lab functional study must be carried out.
17) Line 322, replace "exited" with "were".
18) Line 323, add "to" after "...genes related".
19) Line 324, replace "verified" with "verify".
20) Line 373, explain how the data were used, such as the raw data or the trimmed data were considered for analysis, what was the cucumber genome version used for mapping (if used), and what software was selected for gene counts (if used).
21) Line 382, what was the incubation time for photosynthetic pigment extraction? Provide a reference.
22) Line 386, was there any DNase treatment after nucleic acid extraction?
23) In the method section, the plant growth condition is missing.
24) Line 399, replace "a quite a significant difference" with "different".
Therefore, based on the major concern of RNASeq analysis, the role of these two gene sets in the (a)biotic stress as mentioned in the conclusions (lines 400-403) is inconclusive.
Author Response
Please see the attachment, thank you!

Round 2
Reviewer 2 Report
I thank the authors to apply the requested edits properly and accordingly.
There are a few errors, but after editing those, the manuscript could be accepted for publication.
1) Line 475, if the methods in reference #40 (as the remaining RNASeq study) is the methodology used for the plant growth condition in the current study, "two three-leaf stage" should be replaced with "two true-leaf stage".
2) In table S3, replace "Identify" with "Identity".
3) Line 464, remove "were used for" for clarity of the sentence.
Good luck!
Author Response
Please see the attachment, thank you!
